# Ultrasound combined with urokinase under key-shaped bone window enhances blood clot lysis in an in vitro model of spontaneous intracerebral hemorrhage

Lei Xu[1,2◉], Qiang Yang[1,2◉], Jian Gong[1,2], Jia Wang[1,2], Weiming Xiong[1,2], Liu Liu[1,2], Yang Liu[1,2], Weiduo Zhou[1,2], Chao Sun[1,2], Yidan Liang[1,2], Yanglingxi Wang[1,2], Yi Xiang[1,2], Yongbing Deng[1,2], Min Cui[1,2]*

1 Department of Neurosurgery, Chongqing Emergency Medical Center, Chongqing University Central Hospital, Chongqing, 400010, China, 2 Chongqing Key Laboratory of Emergency Medicine, Chongqing Emergency Medical Center, Chongqing University Central Hospital, Chongqing, 400010, China

◉ These authors contributed equally to this work.
* tocuimin@foxmail.com

**Data Availability Statement:** Data are available from the corresponding author (tocuimin@foxmail.

## Abstract

### Objective

Minimally invasive surgery for spontaneous intracerebral hemorrhage is impeded by inadequate lysis of the target blood clot. Ultrasound is thought to expedite intravascular thrombolysis, thereby facilitating vascular recanalization. However, the impact of ultrasound on intracerebral blood clot lysis remains uncertain. This study aimed to explore the feasibility of combining ultrasound with urokinase to enhance blood clot lysis in an in vitro model of spontaneous intracerebral hemorrhage.

### Methods

The blood clots were divided into four groups: control group, ultrasound group, urokinase group, and ultrasound + urokinase group. Using our experimental setup, which included a key-shaped bone window, we simulated a minimally invasive puncture and drainage procedure for spontaneous intracerebral hemorrhage. The blood clot was then irradiated using ultrasound. Blood clot lysis was assessed by weighing the blood clot before and after the experiment. Potential adverse effects were evaluated by measuring the temperature variation around the blood clot in the ultrasound + urokinase group.

### Results

A total of 40 blood clots were observed, with 10 in each experimental group. The blood clot lysis rate in the ultrasound group, urokinase group, and ultrasound + urokinase group (24.83 ± 4.67%, 47.85 ± 7.09%, 61.13 ± 4.06%) was significantly higher than that in the control group (16.11 ± 3.42%) ($p = 0.02$, $p < 0.001$, $p < 0.001$). The blood clot lysis rate in the ultrasound + urokinase group (61.13 ± 4.06%) was significantly higher than that in the ultrasound group (24.83 ± 4.67%) ($p < 0.001$) or urokinase group (47.85 ± 7.09%) ($p < 0.001$). In the

com) and an institutional representative (hml2014@163.com) for researchers.

**Funding:** The author(s) received no specific funding for this work.

**Competing interests:** The authors have declared that no competing interests exist.

ultrasound + urokinase group, the mean increase in temperature around the blood clot was $0.26 \pm 0.15°C$, with a maximum increase of $0.38 \pm 0.09°C$. There was no significant difference in the increase in temperature regarding the main effect of time interval (F = 0.705, $p$ = 0.620), the main effect of distance (F = 0.788, $p$ = 0.563), or the multiplication interaction between time interval and distance (F = 1.100, $p$ = 0.342).

## Conclusions

Our study provides evidence supporting the enhancement of blood clot lysis in an in vitro model of spontaneous intracerebral hemorrhage through the combined use of ultrasound and urokinase. Further animal experiments are necessary to validate the experimental methods and results.

## Introduction

Although the incidence of spontaneous intracerebral hemorrhage is lower than that of ischemic stroke, it results in more severe neurological damage, higher mortality rates, and poorer prognoses [1]. Recent advancements in surgical instruments and techniques have created conditions conducive to precise minimally invasive puncture surgery for treating spontaneous intracerebral hemorrhage. While the STICH trials [2,3] suggested that surgery did not lead to a better prognosis for patients, findings from the MISTIE trials [4–6] suggested that minimally invasive surgery improved functional outcomes for patients with a residual hematoma volume of 15 ml or less. The latest ENRICH trial [7] was the first to indicate that surgical removal of intracerebral hematoma within 24 hours of symptom onset could enhance functional outcomes in patients with spontaneous intracerebral hemorrhage. Consequently, early reduction of hematoma volume may play an important role in enhancing the efficacy of surgical interventions.

Intra-hematoma injection of thrombolytic drugs, such as alteplase or urokinase, has become a widely employed clinical treatment for patients with spontaneous intracerebral hemorrhage [1]. However, the effectiveness of hematoma lysis solely through the injection of thrombolytic drugs is limited due to the localized diffusion around the drainage tube tip and the constraints imposed by the indwelling time, volume, and frequency of drug injections. In the MISTIE III trial [6], the proportion of postoperative residual hematoma volumes > 15 ml was 43% (108/250), indicating the need to address the challenge of accelerating hematoma lysis to facilitate drainage.

The combined use of ultrasound with thrombolytic drugs has shown significant efficacy in clinical applications such as middle cerebral artery thrombosis, peripheral deep vein thrombosis, and pulmonary artery thrombosis [8–11]. However, it remains unknown whether this combination can enhance intracerebral hematoma lysis in patients with spontaneous intracerebral hemorrhage. In an in vitro experiment [12], researchers implanted an endosonography probe into blood clots in vitro and found that ultrasound combined with thrombolytic drugs dissolved clots more effectively compared with the use of ultrasound or thrombolytic drugs alone. However, the placement of an ultrasound probe into the cerebrum is invasive [13]. If extracorporeal ultrasound proves effective in enhancing intracerebral hematoma lysis, it could significantly strengthen its clinical utility.

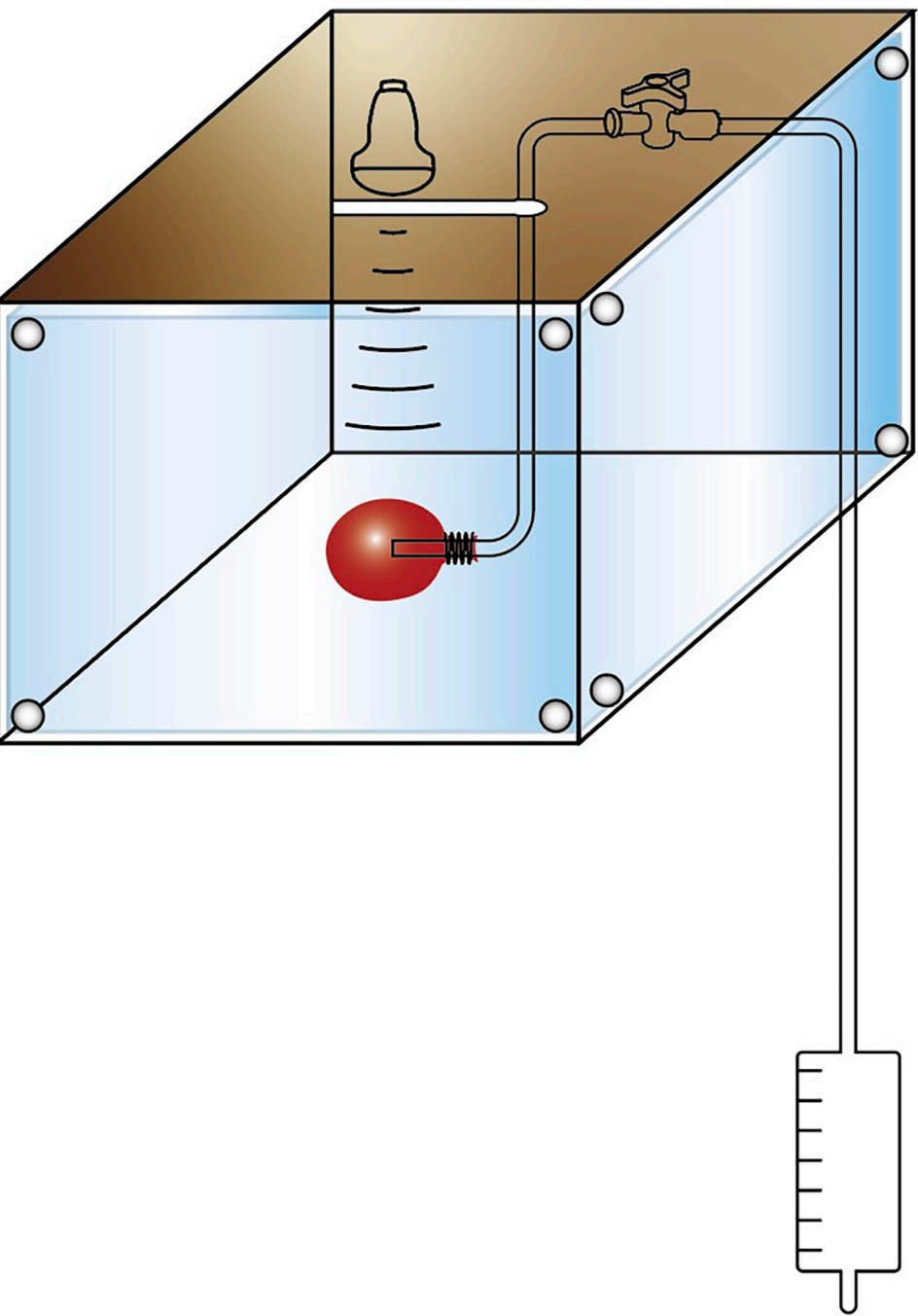

**Fig 1. Experimental setup for ultrasound combined with urokinase to enhance blood clot lysis in vitro.** The 37°C thermostatic water bath simulates the cranial cavity in Fig 2, with the white, key-shaped defect area of the lid of the water bath representing the key-shaped bone window of the skull in Fig 2B, and the red balloon simulating the intracerebral hematoma. The complete set of experimental devices simulates the puncture and drainage of the intracerebral hematoma and continuous ultrasound irradiation of the hematoma, as shown in Fig 2C.

Conventional transcranial ultrasound is impeded by the skull, resulting in a substantial loss of ultrasound energy [14]. Considering these challenges and the need for surgical feasibility, we developed an experimental device (**Fig 1**) based on a key-shaped bone window, as per

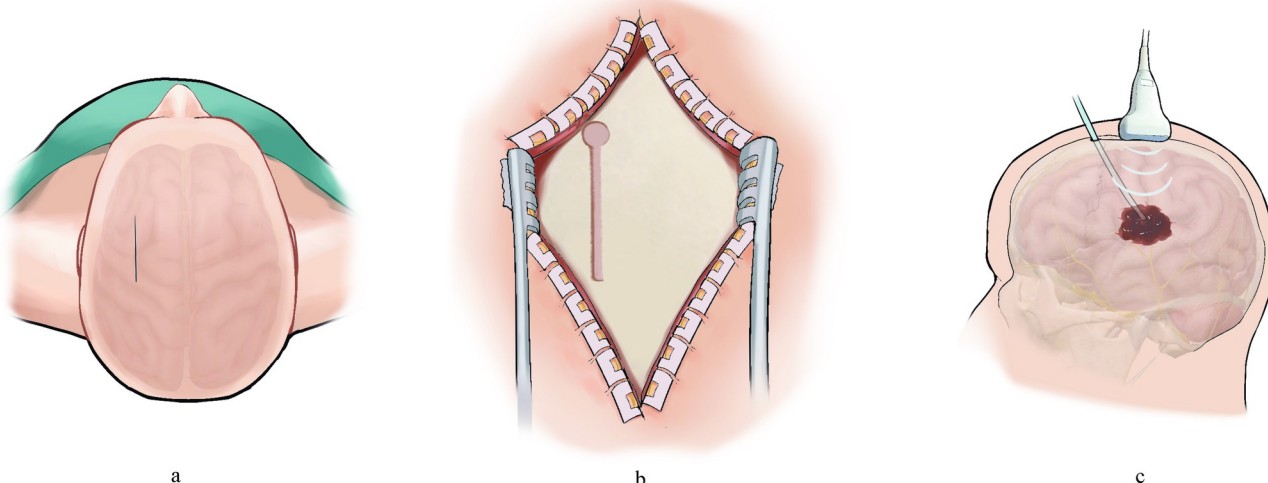

**Fig 2. Hypothetical diagram illustrating the use of ultrasound in combination with urokinase to enhance intracerebral hematoma lysis.** (a) Surgical incision. (b) Scalp pulled back and a key-shaped bone window cut into the skull (The key-shaped bone window consists of two parts: The bone pore and the bone gap. The diameter of the bone pore is 1.5 cm, determined by the surgical drill size. The width of the bone gap is > 0.5 mm, determined by the ultrasonic wavelength. The length of the bone gap can be adjusted according to the sagittal diameter of the hematoma.). (c) Hematoma punctured, drainage tube placed, and continuous ultrasound irradiation applies through the key-shaped bone window.

previous studies [12,13,15,16]. This device simulated minimally invasive puncture and drainage surgery for spontaneous intracerebral hemorrhage, while using ultrasound to irradiate the blood clot (**Fig 2**). The objective of this study is to investigate the feasibility of using ultrasound combined with urokinase to enhance blood clot lysis in vitro.

## Materials and methods

### Experimental equipment

The following experimental equipment was used in this study. The color ultrasound diagnostic system (Wisonic Clivia 90, Shenzhen Huasheng Medical Technology Co., Ltd., China): pulsed-wave mode, frequency (4.5 MHz), power (91 mW), mechanical index (1.8) and duration (1 hour). Thermostatic water bath (JOANLAB, model BHS, China). Temperature measuring instrument (YOWEXA, model YET-620L, China). Scale (KTURE high-precision scale, China). Drainage tube (Weihai Chuangshi Medical Technology Co., Ltd., model NSYL-II [external diameter 4 mm, 2 side holes], China).

### Preparation and grouping of blood clots

Following previous studies [17,18], a solution of 600 µl 5% calcium chloride (Xiamen Haibiao Technology Co., Ltd., China) was combined with 30 ml of fresh sterile anticoagulant bovine whole blood (Zhengzhou Pingrui Biotechnology Co., Ltd., China) in a balloon. The mixture was thoroughly blended, and the balloon was securely sealed using sutures after completion. Subsequently, the sealed balloon was immersed in a 37°C thermostatic water bath, with its center positioned 8 cm from the water surface. The incubation period lasted for 3 hours to facilitate the formation of a stable blood clot (**Fig 3**).

With reference to previous literature [15] (residual hematoma weight as the main outcome), the sample size was calculated using PASS 15.0 software to be 16 (power = 90%, α = 0.05, dropout rate = 20%). However, since our study was an exploratory experiment, and

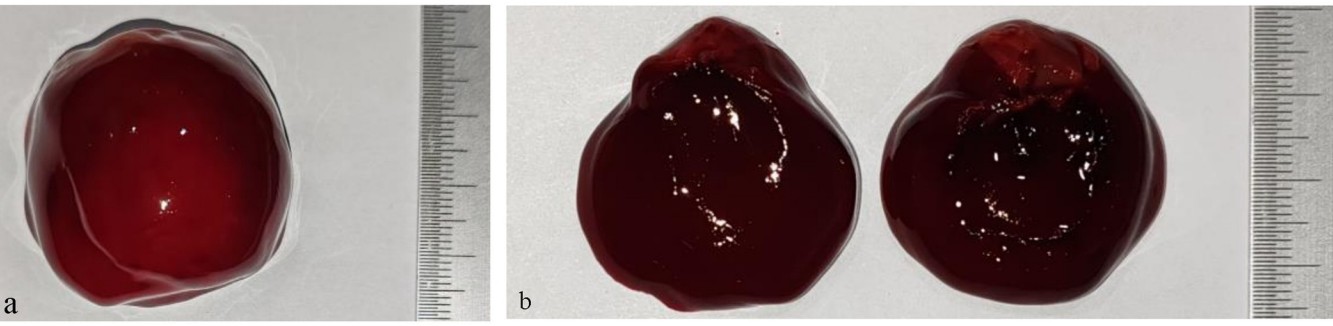

**Fig 3. Preparation of a blood clot.** The smallest scale on the ruler is 0.5mm. (a) Formation of a blood clot inside a balloon. (b) Median dissection of the blood clot.

considering the differences between our experimental methods and the reference, we set the sample size to 40.

A total of 40 blood clots were prepared using the methods described above. The blood clots were then categorized into four groups—control group, ultrasound group, urokinase group, and ultrasound + urokinase group—based on the random number table method. Each group consisted of 10 blood clots.

## Medications for injection

Urokinase (Wuhan Renfu Pharmaceutical Co., Ltd., 100,000 IU, China) was prepared into a solution with 10,000 IU/ml using normal saline (Southwest Pharmaceutical Co., Ltd., 0.9% concentration, China). The prepared solution was stored in a 5˚C refrigerator, and it was reformulated if kept for more than 24 hours. For the control and ultrasound groups, 3 ml of normal saline was injected. In the urokinase group and ultrasound + urokinase group, 20,000 IU of urokinase was injected, followed by flushing the tube with 1 ml of normal saline (the drainage tube's cavity volume was 1 ml). The volume of the transcatheter injection solution into each blood clot was standardized to 3 ml.

## Experimental procedure

Following the preparation of the blood clot, the balloon containing the blood clot was weighed to determine the pre-treatment weight, after removing surface water from the balloon. The balloon was then securely ligated with sutures after placing a drainage tube at the center of the blood clot. Subsequently, the balloon was positioned in a 37˚C thermostatic water bath, with the center of the blood clot situated 8 cm under the water surface. This arrangement aimed to simulate the distance between the scalp and the intracerebral hematoma in the basal ganglia on the sagittal plane. The other end of the drainage tube was connected to a drainage bag, positioned 20 cm below the blood clot, allowing gravity to facilitate the drainage of the liquefied blood clot (**Fig 1**).

For the control group, a volume of 3 ml normal saline was injected into the blood clot through a three-way valve, and the valve was closed for a duration of 1 hour. In the ultrasound group, a similar volume of normal saline was injected into the blood clot through the three-way valve, the valve was then closed, and the tip of the drainage tube was positioned through the key-shaped bone window using ultrasound, followed by continuous irradiation for 1 hour. In the urokinase group, 20,000 IU of urokinase was initially injected into the blood clot through the three-way valve, followed by the injection of 1 ml of normal saline to flush the drainage tube. The three-way valve was closed for 1 hour. In the ultrasound + urokinase

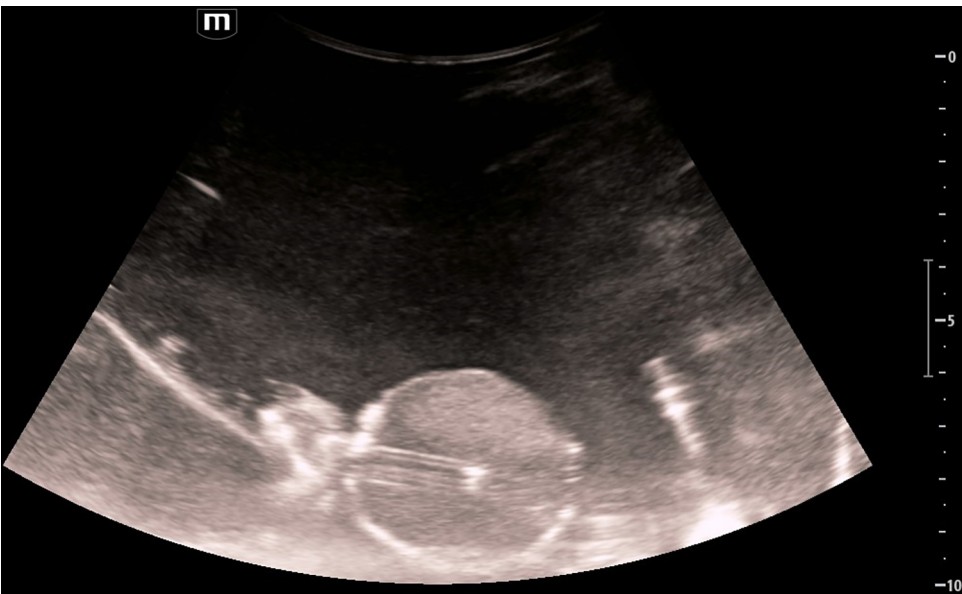

**Fig 4. Blood clot in a thermostatic water bath irradiated with ultrasound.** The smallest scale on the ruler is 5mm.

group, 20,000 IU of urokinase was injected into the blood clot through the three-way valve, followed by 1 ml of normal saline to flush the drainage tube. The three-way valve was then closed, and the tip of the drainage tube was positioned through the key-shaped bone window using ultrasound, with continuous irradiation for 1 hour (**Fig 4**).

After the above treatments, the three-way valve was opened to drain the liquefied blood clot in all experimental groups. The valve was closed after continuous drainage for 1 hour. The drainage tube and sutures were removed, and any remaining water on the balloon's surface was dried with absorbent paper. The balloon and the residual blood clot were then weighed to obtain the post-treatment weight.

Blood clot lysis rate was calculated using the formula: Blood clot lysis rate = (pre-treatment weight—post-treatment weight) / (pre-treatment weight—balloon weight) x 100%. Pre-treatment weight: this was determined after the formation of the blood clot. The weight of the balloon and the blood clot was measured after removing surface water from the balloon using absorbent paper. Post-treatment weight: after the experiment, the drainage tube and suture were removed, and any water on the surface of the balloon was removed with absorbent paper. The weight of the balloon and the residual blood clot was then measured.

Key-shaped bone window: the key-shaped bone window consists of two parts: the bone pore and the bone gap (**Fig 2B**). The diameter of the bone pore was 1.5 cm, determined by the surgical drill size. The width of the bone gap was > 0.5 mm, determined by the ultrasonic wavelength. The length of the bone gap can be adjusted according to the sagittal diameter of the hematoma.

## Temperature measurement in the ultrasound + urokinase group

In the Ultrasound + Urokinase group, the temperature of the balloon surface and at distances of 2 cm, 4 cm, 6 cm, 8 cm, and 10 cm away from the balloon surface was recorded using a temperature measuring instrument. Measurements were taken at 10-minute intervals starting from the initiation of ultrasound irradiation.

## Statistical analysis

Continuous variables were presented as mean ± standard deviation (SD). Group comparisons were executed using one-way analysis of variance (ANOVA) or two-way repeated measures ANOVA for continuous variables. Post-hoc multiple comparisons were adjusted using the Bonferroni method to account for potential Type I errors. All statistical tests were two-tailed, and a significance level of $P < 0.05$ was adopted for determining statistical significance. Statistical analyses were carried out utilizing SPSS 25.0 (IBM).

## Results

### Blood clot lysis rate

A total of 40 blood clots were observed, with 10 in each experimental group. The pre-treatment weights of blood clots in the control group, ultrasound group, urokinase group, and ultrasound + urokinase group were 31.05 ± 0.28 g, 31.15 ± 0.32 g, 31.34 ± 0.47 g, and 31.14 ± 0.40 g, respectively. No significant difference in the pre-treatment weight of blood clots was observed among the four groups ($p = 0.381$).

Post-treatment, the blood clot lysis rate in the ultrasound group, urokinase group, and ultrasound + urokinase group (24.83 ± 4.67%, 47.85 ± 7.09%, 61.13 ± 4.06%) was significantly higher compared to the control group (16.11 ± 3.42%) ($p = 0.002$, $p < 0.001$, $p < 0.001$). Moreover, the blood clot lysis rate in the ultrasound + urokinase group (61.13 ± 4.06%) was significantly higher than that in the ultrasound group (24.83 ± 4.67%) ($p < 0.001$) or urokinase group (47.85 ± 7.09%) ($p < 0.001$).

### Temperature variation in the ultrasound + urokinase group

During the one hour of ultrasound irradiation, the temperature of the balloon surface and within a 10 cm vicinity was measured at different time points. The average temperature increase was 0.26 ± 0.15˚C, with a minimum of 0.18 ± 0.13˚C and a maximum of 0.38 ± 0.09˚C.

The analysis utilizing two-way repeated measures ANOVA revealed no significant difference in temperature increase concerning the main effect of time interval (F = 0.705, $p = 0.620$), the main effect of distance (F = 0.788, $p = 0.563$), or the multiplication interaction of time interval and distance (F = 1.100, $p = 0.342$) (Table 1).

**Table 1. Temperature variation in the ultrasound + urokinase group.**

| Time Interval (min) | Temperature of Different Distance (˚C) | | | | | | F-value | P-value[a] |
| --- | --- | --- | --- | --- | --- | --- | --- | --- |
| | 0cm | 2cm | 4cm | 6cm | 8cm | 10cm | | |
| **10** | 0.20±0.16 | 0.25±0.18 | 0.24±0.18 | 0.29±0.13 | 0.26±0.14 | 0.30±0.14 | | |
| **20** | 0.26±0.16 | 0.23±0.12 | 0.24±0.17 | 0.29±0.15 | 0.22±0.17 | 0.26±0.16 | | |
| **30** | 0.25±0.17 | 0.21±0.15 | 0.28±0.10 | 0.24±0.14 | 0.20±0.15 | 0.26±0.15 | | |
| **40** | 0.34±0.13 | 0.27±0.19 | 0.19±0.18 | 0.20±0.16 | 0.18±0.13 | 0.27±0.12 | | |
| **50** | 0.23±0.14 | 0.24±0.17 | 0.21±0.15 | 0.37±0.11 | 0.31±0.16 | 0.29±0.16 | | |
| **60** | 0.29±0.12 | 0.28±0.15 | 0.24±0.15 | 0.29±0.17 | 0.38±0.09 | 0.20±0.11 | | |
| **Main effect of time interval** | | | | | | | 0.705 | 0.620 |
| **Main effect of distance** | | | | | | | 0.788 | 0.563 |
| **Interaction** | | | | | | | 1.100 | 0.342 |

[a] $p$ values were calculated using the two-way repeated measures analysis of variance.

## Discussion

This study simulated minimally invasive puncture surgery in patients with spontaneous intra-cerebral hemorrhage. The observed blood clot lysis rates in the ultrasound group, urokinase group, and ultrasound + urokinase group ($24.83 \pm 4.67\%$, $47.85 \pm 7.09\%$, $61.13 \pm 4.06\%$) were significantly higher than those in the control group ($16.11 \pm 3.42\%$) ($p = 0.002$, $p < 0.001$, $p < 0.001$). Moreover, the blood clot lysis rate in the ultrasound + urokinase group ($61.13 \pm 4.06\%$) was markedly higher compared to the ultrasound group ($24.83 \pm 4.67\%$) ($p < 0.001$) or the urokinase group ($47.85 \pm 7.09\%$) ($p < 0.001$). In the ultrasound + urokinase group, the observed mean increase in temperature around the blood clot was $0.26 \pm 0.15°C$, with a maximum increase of $0.38 \pm 0.09°C$. Importantly, there was no statistically significant difference in the temperature increase concerning the main effect of time interval (F = 0.705, $p = 0.620$), the main effect of distance (F = 0.788, $p = 0.563$), or the multiplication interaction between time interval and distance (F = 1.100, $p = 0.342$).

The widely accepted mechanism of sonothrombolysis involves acoustic radiation pressure and ultrasound-induced stable cavitation and inertial cavitation. These processes generate microflow and microjets, leading to the temporary release of the fibrin clot. This phenomenon serves to enhance the diffusion of thrombolytic drugs within the blood clot, ultimately improving the therapeutic efficacy of thrombolytic agents. The synergistic effect of ultrasound contributes to faster and more effective thrombolysis. These mechanisms indicate the potential of ultrasound as a valuable adjunct to thrombolytic therapy [19–24].

In peripheral blood vessels, sonothrombolysis had been demonstrated to be both safe and effective. A meta-analysis focused on peripheral deep vein thrombosis encompassed 18 studies with a total of 597 patients [10]. The overall success rate of ultrasound-accelerated catheter-directed thrombolysis (USACDT) in patients with deep vein thrombosis was 87.8% (18 studies, 95% confidence interval [CI]: 83.1–91.3). Notably, the success rate of USACDT was significantly higher than that of conventional catheter thrombolysis (7 studies, odds ratio [OR]: 2.96, 95% CI: 1.69–5.16, $P < 0.01$). Importantly, there was no significant difference in the total complication rates between the two treatments.

In a case report [11] discussing pulmonary embolism, ultrasonographic accelerated thrombolytic therapy was employed in three patients with acute pulmonary embolism who had contraindications to systemic thrombolysis. All three patients exhibited significant clinical and hemodynamic improvements in the short term, achieving thrombolysis and rapid decreases in systolic and mean pulmonary artery pressure.

Numerous clinical studies had explored the role of ultrasound in promoting the lysis of intracranial arterial thrombosis. In a meta-analysis of acute ischemic stroke encompassing seven randomized controlled clinical trials [9], patients diagnosed with large vessel occlusion were randomly assigned to either an ultrasound thrombolysis group (138 cases) or an intravenous thrombolysis alone group (134 cases). The probability of complete recanalization was found to be increased in the ultrasound group (40.3% vs. 22.4%, Adjusted OR: 2.33, 95% CI: 1.02–5.34). Importantly, there was no significant difference in intracranial hemorrhage between the two groups. These findings provided a safe and feasible basis for the clinical application of ultrasound in dissolving both peripheral and intracranial intravascular thrombosis.

The use of ultrasound for the assisted lysis of intracerebral hematoma is infrequently reported and is primarily limited to in vitro studies. In one such in vitro study [13], human blood clots (volume: 25 ml) were categorized into six different treatment groups: control group, alteplase group, one doppler probe group, two doppler probes group, one doppler probe + alteplase group, and two doppler probes + alteplase group. Notably, the group with two doppler probes plus alteplase achieved the most effective blood clot lysis ($36.3 \pm 4.4\%$).

Temperature measurements were taken at distances of 0–7 cm from the ultrasound probe, revealing a maximum temperature increase of 0.17 ± 0.07˚C. The authors pointed out that the radiation of ultrasound energy was significantly reduced in the presence of the skull. Although bilateral transcranial Doppler (TCD) can improve the attenuation of ultrasound energy by the skull, challenges remain in accurately placing two TCD probes completely opposite each other and precisely locating the hematoma with the TCD probe.

To address these challenges, we designed a key-shaped bone window for use in clinical minimally invasive surgery. This design allowed for implementation under a straight scalp incision, effectively mitigating the influence of the skull on ultrasound. Importantly, the dura was not opened after cutting the key-shaped bone window in the skull, minimizing damage to brain tissue. Ultrasound images can accurately locate the hematoma, enhancing clinical applicability. While the maximum temperature increase in this study was higher than that reported in previous studies, the observed temperature increase range in this study was deemed clinically acceptable.

In addition to in vitro studies, a small clinical trial [25] included eight patients with spontaneous intracerebral hemorrhage (three intraventricular and five intracerebral). The study involved placing a drainage catheter and an ultrasound microcatheter into the hematoma and injecting alteplase. The mean percent volume reduction of intracerebral hemorrhage and intraventricular hemorrhage was 59 ± 5% and 45.1 ± 13%, respectively. No intracranial infections or significant rebleeding events occurred. A comparison between patients treated with alteplase alone and those treated with ultrasound plus alteplase revealed that blood clots dissolved faster in the latter group. While this study presented alternative methods for ultrasound-assisted lysis of intracerebral blood clots, it acknowledged the limitations associated with ultrasound microcatheter instrument and the invasive nature of ultrasound microcatheter placement in the brain, making its clinical practicality and safety lower compared to extracorporeal ultrasound.

## Limitations

The study has several limitations that should be considered. First, the use of a 37˚C thermostatic water bath may not fully replicate the complex pathophysiological environment present in the human brain. The in vitro nature of the experimental setup may not capture all the dynamic factors involved in spontaneous intracerebral hemorrhage in living organisms. Second, patients with spontaneous intracerebral hemorrhage often present with underlying diseases such as hyperlipidemia and diabetes. The use of bovine whole blood may not entirely replicate the diverse blood components and biochemical profile found in patients with these comorbidities. Third, while temperature measurements around the blood clot were taken, the study was unable to assess secondary brain damage, including the potential for rebleeding and edema around the blood clot. The limitations of in vitro experiments constrain the ability to replicate the full spectrum of clinical scenarios and outcomes associated with spontaneous intracerebral hemorrhage. Fourth, since only one ultrasonic frequency and duration were used in our experiment, further studies are needed to determine the most suitable frequency and duration for promoting blood clot lysis.

These limitations highlight the need for caution when extrapolating the findings of the study to the clinical setting. Future research involving animal models or clinical trials would provide a more comprehensive understanding of the potential applications and limitations of ultrasound-assisted blood clot lysis in patients with spontaneous intracerebral hemorrhage.

## Conclusions

These findings suggest that the combined application of ultrasound and urokinase effectively enhances blood clot lysis in an in vitro model of spontaneous intracerebral hemorrhage. The

observed temperature variations indicate a controlled and manageable thermal effect, supporting the safety and feasibility of this combined approach. However, it is crucial to acknowledge that these results are based on in vitro experiments, and further validation through animal studies and clinical trials is essential to ascertain the practical applicability and safety of this therapeutic strategy in a clinical setting.

## Supporting information

**S1 Checklist.**
(DOCX)

## Author Contributions

**Conceptualization:** Lei Xu, Min Cui.

**Data curation:** Qiang Yang, Jia Wang, Liu Liu, Chao Sun, Yanglingxi Wang.

**Formal analysis:** Qiang Yang.

**Investigation:** Lei Xu.

**Methodology:** Qiang Yang.

**Project administration:** Lei Xu.

**Software:** Jia Wang, Liu Liu, Chao Sun, Yanglingxi Wang.

**Supervision:** Yongbing Deng, Min Cui.

**Writing – original draft:** Jian Gong, Weiming Xiong, Yang Liu.

**Writing – review & editing:** Weiduo Zhou, Yidan Liang, Yi Xiang.

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
