## [Decision Letter · Decision Letter 0]

16 Apr 2024

PONE-D-24-11039Ultrasound Combined with Urokinase Under Key-Shaped Bone Window Enhances Blood Clot Lysis in an In Vitro Model of Spontaneous Intracerebral HemorrhagePLOS ONE

Dear Dr. cui,

Thank you for submitting your manuscript to PLOS ONE. After careful consideration, we feel that it has merit but does not fully meet PLOS ONE’s publication criteria as it currently stands. Therefore, we invite you to submit a revised version of the manuscript that addresses the points raised during the review process.

We look forward to receiving your revised manuscript.

Kind regards,

Atakan Orscelik

Academic Editor

PLOS ONE

2. In this instance it seems there may be acceptable restrictions in place that prevent the public sharing of your minimal data. However, in line with our goal of ensuring long-term data availability to all interested researchers, PLOS’ Data Policy states that authors cannot be the sole named individuals responsible for ensuring data access (http://journals.plos.org/plosone/s/data-availability#loc-acceptable-data-sharing-methods).

Reviewers' comments:

Reviewer's Responses to Questions

**Comments to the Author**

1. Is the manuscript technically sound, and do the data support the conclusions?

Reviewer #1: Yes

Reviewer #2: Yes

2. Has the statistical analysis been performed appropriately and rigorously? 

Reviewer #1: Yes

Reviewer #2: I Don't Know

3. Have the authors made all data underlying the findings in their manuscript fully available?

Reviewer #1: Yes

Reviewer #2: Yes

4. Is the manuscript presented in an intelligible fashion and written in standard English?

Reviewer #1: Yes

Reviewer #2: Yes

5. Review Comments to the Author

Reviewer #1: nice work

although this work need further validation by animal studies

it would be great if you mention the exact strength(other characteristics like frequency, power, duration) of ultrasound waves used for clot lysis.

Reviewer #2: Cui  et al are presenting a study aiming to evaluate the role of ultrasound combined with urokinase to enhance blood clot lysis in an in vitro model of spontaneous intracerebral hemorrhage. The manuscript is of scientific interest  and contains data that can easily be translated to the clinical field.  My major concern is the absence of discussion about the optimal frequency/duration and the absence  of an explanation of the study sample calculation.  I also have the following comments: 

1.    "Spontaneous intracerebral hemorrhage is classified as a subtype of stroke"- this information is trivial. Please remove this sentence

2. "Reference 12 reported that ultrasound catheters combined with alteplase enhancedblood clot lysis in vitro" - please clarify the meaning of this sentence3."However, due to the invasive nature of ultrasound catheters, theirclinical application is currently restricted to peripheral vascular lesions" - provide the appropriate bibliographic support 

4. It is very hard to understand the meaning of figure 2 in the context of this study (in vitro experiment) - it is merely speculative. 

5. The manuscript needs english revision - some sentences are very difficult to understand.

6. PLOS authors have the option to publish the peer review history of their article (what does this mean?). If published, this will include your full peer review and any attached files.

Reviewer #1: **Yes: **Dr Anand Kumar

MD, DM

Assistant Professor, IMS BHU, Varanasi, India

Reviewer #2: No

---

## [Author Response · Author response to Decision Letter 0]

24 Apr 2024

Reviewer #1: nice work

although this work needs further validation by animal studies, it would be great if you mention the exact strength (other characteristics like frequency, power, duration) of ultrasound waves used for clot lysis.

Reply: Thank you for your comments and suggestions. We have provided the relevant characteristics of ultrasound wave. (page 5)

Reviewer #2: Cui et al are presenting a study aiming to evaluate the role of ultrasound combined with urokinase to enhance blood clot lysis in an in vitro model of spontaneous intracerebral hemorrhage. The manuscript is of scientific interest and contains data that can easily be translated to the clinical field. My major concern is the absence of discussion about the optimal frequency/duration and the absence of an explanation of the study sample calculation. 

Reply: Thank you for your comments and suggestions. 

We have provided an explanation of the study sample calculation. (page 5/6)

Since we only used one frequency and duration of ultrasound waves, we do not have sufficient data to support the discussion of frequency and duration. We have included this as a limitation of our study. We appreciate your understanding. (page 13)

I also have the following comments: 

1. "Spontaneous intracerebral hemorrhage is classified as a subtype of stroke"- this information is trivial. Please remove this sentence.

Reply: Thank you for your comments and suggestions. We have made the appropriate revisions. (page 3)

2. "Reference 12 reported that ultrasound catheters combined with alteplase enhanced blood clot lysis in vitro" - please clarify the meaning of this sentence.

Reply: Thank you for your comments and suggestions. We have made the appropriate revisions. (page 4)

3. "However, due to the invasive nature of ultrasound catheters, their clinical application is currently restricted to peripheral vascular lesions" - provide the appropriate bibliographic support.

Reply: Thank you for your comments and suggestions. We have made the appropriate revisions. (page 4)

4. It is very hard to understand the meaning of figure 2 in the context of this study (in vitro experiment) - it is merely speculative.

Reply: Thank you for your comments and suggestions. We have revised the legends for Figure 1 and Figure 2. (page 16)

5. The manuscript needs English revision - some sentences are very difficult to understand.

Reply: Thank you for your comments and suggestions. We have made the appropriate revisions.

---

## [Decision Letter · Decision Letter 1]

13 May 2024

Ultrasound Combined with Urokinase Under Key-Shaped Bone Window Enhances Blood Clot Lysis in an In Vitro Model of Spontaneous Intracerebral Hemorrhage

PONE-D-24-11039R1

Dear Dr. Min Cui,

We’re pleased to inform you that your manuscript has been judged scientifically suitable for publication and will be formally accepted for publication once it meets all outstanding technical requirements.

Kind regards,

Atakan Orscelik

Academic Editor

PLOS ONE

Reviewers' comments:

Reviewer's Responses to Questions

**Comments to the Author**

1. If the authors have adequately addressed your comments raised in a previous round of review and you feel that this manuscript is now acceptable for publication, you may indicate that here to bypass the “Comments to the Author” section, enter your conflict of interest statement in the “Confidential to Editor” section, and submit your "Accept" recommendation.

Reviewer #2: All comments have been addressed

Reviewer #3: All comments have been addressed

2. Is the manuscript technically sound, and do the data support the conclusions?

Reviewer #2: Yes

Reviewer #3: Partly

3. Has the statistical analysis been performed appropriately and rigorously? 

Reviewer #2: Yes

Reviewer #3: Yes

4. Have the authors made all data underlying the findings in their manuscript fully available?

Reviewer #2: Yes

Reviewer #3: Yes

5. Is the manuscript presented in an intelligible fashion and written in standard English?

Reviewer #2: Yes

Reviewer #3: Yes

6. Review Comments to the Author

Reviewer #2: The authors have adressed all my comments. I do not have any further comments. The manuscript os ready for publication in the current form.

Reviewer #3: (No Response)

7. PLOS authors have the option to publish the peer review history of their article (what does this mean?). If published, this will include your full peer review and any attached files.

Reviewer #2: No

Reviewer #3: No

---

## [Editor Report · Acceptance letter]

19 May 2024

PONE-D-24-11039R1 

PLOS ONE

Dear Dr. cui, 

I'm pleased to inform you that your manuscript has been deemed suitable for publication in PLOS ONE. Congratulations! Your manuscript is now being handed over to our production team.

Kind regards, 

on behalf of

Dr. Atakan Orscelik 

Academic Editor

PLOS ONE